# Intraguild Interactions of Three Biological Control Agents of the Fall Armyworm *Spodoptera frugiperda* (JE Smith) in Florida

**DOI:** 10.3390/insects13090815

**Published:** 2022-09-07

**Authors:** Jermaine D. Perier, Muhammad Haseeb, Lambert H. B. Kanga, Robert L. Meagher, Jesusa C. Legaspi

**Affiliations:** 1Center for Biological Control, College of Agriculture and Food Sciences, Florida A&M University, Tallahassee, FL 32307, USA; 2Department of Entomology, University of Georgia, Tifton, GA 31794, USA; 3Insect Behavior and Biocontrol Research Unit, ARS-CMAVE, USDA, Gainesville, FL 32608, USA; 4Insect Behavior and Biocontrol Research Unit, Center for Medical, Agricultural and Veterinary Entomology, Agricultural Research Service, United States Department of Agriculture, Tallahassee, FL 32308, USA

**Keywords:** pest, maize, insect behavior, predator competition, parasitoid, IPM

## Abstract

**Simple Summary:**

*Spodoptera frugiperda* is a native noctuid pest of the Western Hemisphere with a growing global distribution. It is polyphagous by nature and is the primary insect pest attacking maize in Florida. Larval feeding damage has resulted in crop yield losses of up to 20% or more in the United States. In other parts of the world, it causes maize yield losses of between 17–72%. Consequently, regular pesticide applications have led to insecticide-resistant strains of *S. frugiperda*. Herein, we propose using biological control agents as an alternative pest management strategy. Augmentative biological control allows for the strategic release of these agents where they are sparse or absent in open fields. However, insects are complex organisms with an array of intraguild interactions that might contradict such an effort. Therefore, the goal of improving these agents’ use for alternative pest control requires evaluating their interactions as expressed in the same guild.

**Abstract:**

The fall armyworm, *Spodoptera frugiperda* (J.E. Smith) (Lepidoptera: Noctuidae), is a maize pest worldwide. Its host range comprises more than 350 reported plant species, and it is the primary insect pest attacking maize in Florida. Global trade has not only assisted but accelerated its invasion into the Eastern Hemisphere. Regular pesticide use promotes resistance in the species; therefore, there is an urgent need for alternative pest management strategies. This study evaluated the interactions of biological control agents within a similar guild. Some of the reported interactions could potentially lead to the integration of these agents within the same niche to increase biological control efficiency against the fall armyworm. We evaluated three biocontrol agents that are natural enemies of Lepidopteran pests, the true bugs *Podisus maculiventris* and *Euthyrhynchus floridanus* (Hemiptera: Pentatomidae) and a parasitoid, *Cotesia marginiventris* (Hymenoptera: Braconidae). Depending on their intraguild interactions, these agents could potentially be useful for biological control of the fall armyworm. The study investigated these three biocontrol agents and concluded that integrating these agents to control the fall armyworm is a possibility; however, only under certain conditions. Investigations were focused on evaluating the predator–parasitoid and devised pairing interactions. Predator response to prey in a choice or no-choice scenario and choices based on olfaction or other bodily cues were studied under experimental laboratory conditions.

## 1. Introduction

The fall armyworm, *Spodoptera frugiperda* (J.E. Smith, 1797) (Lepidoptera: Noctuidae), is an important pest of crops in the Western Hemisphere, with a regional distribution range from southern Canada to Argentina [1]. In Florida, USA, the fall armyworm is the primary pest attacking the ears of sweet maize (*Zea mays* L.) [2]. Additional reports have also indicated significant economic damage to sorghum (*Sorghum vulgare* Pers.) and turf grass across the United States [3,4,5]. The fall armyworm is polyphagous and feeds on more than 350 plant species globally, with a preference for maize and rice [1,6,7]. Maize production in the United States accounts for 55% of global production [8], with Florida contributing 24% (USD 160 million) of the total U.S. value [9,10]. Given the importance of host crops such as maize globally [8,11] and the recent spread of the fall armyworm to other continents and new U.S. states [12]. this insect has become a pest of global concern [13]. Different countries have reported varying yield losses, such as 17–72% in South America [14,15] and up to 20% in the United States [1]. All larval stages of the fall armyworm cause considerable feeding damage to host plants, especially at high infestation levels [2,16], with most of the damage occurring in the final larval instar [1]. Pest feeding drastically impacts the plant’s growth [17] and reduces ear marketability significantly [18]. Damage mitigation frequently involves the extensive use of insecticides and transgenic crops [19,20]. Consequently, the heavy reliance on insecticides produced insecticide-resistant strains within the species [21], resulting in the further application of larger volumes of insecticides [22]. Ultimately, these actions increase the risk of environmental pollution and potential impacts on non-target organisms [23].

Biological control agents are vital components in many integrated pest management programs due to their natural ability to regulate pest populations [24]. Most natural enemies of the fall armyworm are generalist predators and parasitoids that also attack other pest species in agricultural ecosystems [1]. Including multiple biological control agents could perhaps improve the effectiveness of biological control programs targeting the fall armyworm [13,25]. However, any attempt to integrate multiple generalist natural enemies to control agricultural pests poses concerns about competition and intraguild predation, amongst others [26,27,28]. Moreover, competition amongst biological control agents involves more complex interactions [26]. This study highlights the results of interactions among biological control agents when used simultaneously in a guild. We evaluated two predators, *Podisus maculiventris* (Say, 1832) and *Euthyrhynchus floridanus* (L., 1767) (Hemiptera: Pentatomidae), along with a parasitoid, *Cotesia marginiventris* (Cresson, 1865) (Hymenoptera: Braconidae), for their roles as biological control agents against the fall armyworm. All three biocontrol agents are native to the neotropical regions of the U.S., including Florida [29,30,31]. However, reports on their control efforts against economic pests are not consistent for all three species. This is perhaps a result of the difference in the durations of life cycles (28 days and 60 days, *P. maculiventris* and *E. floridanus* respectively), despite the similarities in size of each instar and adult among the pentatomid predators (head capsule widths: first = 0.6–0.7 mm, second = 0.9 mm, third = 1.2–1.3 mm, fourth = 1.7 mm, fifth = 2.1–2.2 mm instar and adult = 2.3 mm for *P. maculiventris;* and male = 2.3 mm, female = 7.2 mm for *E. floridanus*) [29,32]. The study aimed to understand the response of these three agents, and multiple experiments were conducted under controlled conditions to determine the responses of predators toward the prey. 

## 2. Materials and Methods

The experiment was conducted in a laboratory at the Center for Biological Control, Florida Agricultural and Mechanical University, in Tallahassee, Florida. All colonies were established and reared onsite for the experiments. All colonies were reared for one to three generations before initiating experiments.

### 2.1. Colony Establishment

Colonies of *Podisus maculiventris* and *Euthyrhynchus floridanus* were established using adults and eggs obtained from laboratory cultures (originally from Florida) at the USDA-ARS-CMAVE laboratory in Tallahassee, Florida. Three generations were established for each species before experimentation. Colonies were reared in incubators maintained at 26 ± 2 °C, 65 ± 5% relative humidity and with a 14:10 (L:D) h photoperiod. Additional rearing procedures for both species followed those described by Legaspi and Legaspi [33].

Larvae of the fall armyworm and *Cotesia marginiventris* (henceforth called *Cotesia*) were obtained from existing laboratory cultures (started with Florida field collections) at USDA-ARS-CMAVE in Gainesville, Florida. Due to the different rearing requirements for the life stage of both species, different confinements were prepared.

Adult fall armyworm moths were held in cylindrical rearing mesh cages (20 cm in diameter, 23 cm in height, and 3 mm wire mesh) on shelves in an enclosed insect-rearing room. The tops of the cages were covered with paper towels secured with rubber bands for female moths’ oviposition. Paper towels were collected (and replaced) within 24 h of eggs being laid and placed in labeled bags until larvae were eclosed. Moths were provided a honey/sucrose water solution (10%) in a 60 mL acrylic cup for feeding. Larvae of the fall armyworm were reared in 30 mL cups and provided a store-bought diet (Multiple Species Diet, Southland Products Inc. Lake Village, Arkansas) modified to include 2.87 mL linseed oil/L. All fall armyworm life stages were incubated at 26 ± 2 °C ansd 65 ± 5% R.H. with a 14:10 (L:D) h photoperiod according to the methods described by Perkins [34].

Adult *Cotesia* wasps were transferred to a plastic-framed 20.5 × 20.5 × 20.5 cm container with organdy screen walls and provided a 10% honey/sucrose water solution in a 60 mL cup upon emergence. Wasps were reared at 23 ± 2 °C and 65 ± 5% relative humidity with a 13:11 (L:D) h photoperiod. Mated adults were offered newly molted fall armyworm second or third instar larvae for oviposition. Parasitism of larvae was confirmed visually, and parasitized larvae were held separately in 30 mL cups at 26 ± 2 °C and 65 ± 5% R.H. with a 14:10 (L:D) h photoperiod and fed Multiple Species Diet.

### 2.2. Biological Control Agents’ Interactions

#### 2.2.1. Predator vs. Predator (without Prey)

A no-choice experiment evaluating predation between *Podisus maculiventris* and *Euthyrhynchus floridanus* involved 240 Petri dishes (9 cm diameter, 1.5 cm depth) that were prepared by lining the base dish with filter paper and inserting a cotton ball (presoaked in water). Each Petri dish was a replicate arena for the experiment and contained one adult true bug. Twenty replicate arenas were prepared for each competitive pairing per species (Table 1). In each arena, one adult was paired with either a nymph or an adult (approximately two days after molting) of the other species. Competitive interactions in each arena were then recorded. Food was withheld from all insects used in the experiment 24 h prior to commencement. The arena was observed for 1 h at 25.6 ± 3 °C. Afterward, observations on feeding (if any) were recorded in binary format (1—yes, 0—no).

#### 2.2.2. Predator vs. Predator (with Prey)

The scenario described in Section 2.2.1 was repeated with the addition of the fall armyworm larvae. A third instar fall armyworm larva (one to two days after molting) was added to each Petri dish arena. Only a single pairing was evaluated: adult vs. third instar nymph in the presence of the fall armyworm. All observations were recorded at 20 min intervals. Food was withheld from the predators (pentatomid) for 24 h prior, and observations were made for 1 h at 25.6 ± 3 °C. “Feeding” was defined as the predator actively consuming an insect for more than 40 s. “Probing” (predator inserting beak into an insect) was not considered feeding, especially if it lasted less than 40 s. Feeding behavior data included four responses: pentatomids fed together on the fall armyworm larva; no feeding of any kind occurred; only one predator fed on the fall armyworm larva; one predator fed on the other pentatomid in the arena. Pentatomid preference for the fall armyworm larva was confirmed by establishing a control experiment that involved one adult predator (only) released into an arena with a third instar fall armyworm for 1 h at (26 ± 3 °C). This was repeated for both predators (every ten replicates).

#### 2.2.3. Predators’ Behavior toward Parasitized and Non-Parasitized Larvae

Predatory behavior toward *Cotesia* parasitized fall armyworm larvae were evaluated using parasitized and non-parasitized larvae. In a no-choice experiment, third instar fall armyworm larvae (one to two days after a molt) were used in both trials. In an arena, one pentatomid predator was offered either a parasitized (within 24 h) or a non-parasitized larva as food. Eighty experimental units (90 cm Petri dishes, 1.5 cm depth) were prepared similarly to those described in Section 2.2.1, and food was withheld from the predators for 24 h prior. Third instar predator nymphs were used, and only one predator per experimental unit. Predators were given 20 min to acclimate before the experiment began, after which a random assignment of parasitized and non-parasitized larvae occurred per predator species. Twenty replications were conducted for each larval type. The experimental conditions were 26 ± 3 °C and 55 ± 5% R.H. with a 14:10 (L:D) h photoperiod for 24 h. Larvae mortality (1—dead, 0—alive) was recorded after 24 h. Any surviving parasitized fall armyworm larvae (if any) were placed under rearing conditions (see Section 2.1) to assess *C. marginiventris* survivorship after predation. All larvae were confirmed dead after an additional 24 h exposure.

### 2.3. Predator–Prey Preference

The prey preference of both pentatomid predators was studied using two different techniques. The first was an olfactory method using a Y-tube olfactometer, and the other was a close-contact method using 15 cm Petri dishes as arenas (1.5 cm depth). Both methods were used to evaluate the behavior of the pentatomid when making a prey choice. Each experiment was replicated 20 times per technique per species and lasted 10 min at 26 ± 3 °C. Replicates evaluated second instar nymphs only, one to two days after a molt. Controls for both methods (ten replications each) involved offering a non-parasitized third instar fall armyworm larva only to a second instar nymph. This meant leaving an empty chamber for the olfactometer method, while only the predator and the fall armyworm larva were placed in the Petri dish for the close-contact method.

#### 2.3.1. Olfactometer

Continuous airflow for the olfactometer was maintained using a two-port air delivery system (ADS) (model OLFM-ADS-2AFM2C, Analytical Research Systems, Micanopy, FL, USA) at 0.4 liters per minute (LPM) for all replicates (experimental and controls). Air input and output were maintained at 20 psi. Parasitized and non-parasitized third instar fall armyworm larvae (newly molted, parasitized within 24 h) were loaded into separate air chambers. Before the release of each predator (second instar nymph only) into the Y-tube of the olfactometer, the chambers were allowed two minutes to fill with air. Nymphs were then released into the long leg of the Y-tube. Responses were scored in binary (1—parasitized/0—non-parasitized), with results confirmed if a nymph crawled more than 3 cm into a short arm of the Y-tube within 10 min. The equipment was cleaned thoroughly before repeating the experiment using the other predator species. Cleaning was done by submerging the removable parts of the olfactometer (air chambers and Y-tube) into a non-fragrant soap and lukewarm water bath. Chambers were scrubbed using a soft sponge, while each arm of the Y-tube was cleaned using a 7.62 cm tube brush (polyester bristles). Afterward, both parts were rinsed thoroughly, partially dried with Kimtech^®^ dry wipes (Kimwipes, Neenah, WI, USA) and placed on a rack to air dry. At the time of use, both parts were wiped with Kimwipes to remove any remaining debris.

#### 2.3.2. Petri Dish Arena

Forty 150 mm Petri dishes (depth 1.5 cm) were prepared as described in Section 2.2.1. Additionally, one parasitized (within 24 h) and one non-parasitized fall armyworm larvae were placed at opposite ends of the arena. A second instar predator nymph was released into the designated center of the Petri dish at an equal distance from both larval types (larvae and predators moved freely in the arena, but larva designation was visually confirmed at all times). Observations were made for 10 min after placing the predator, with confirmed prey choice (1—parasitized/0—non-parasitized) being recorded if feeding lasted longer than 40 s to confirm the predator choices. Twenty replicates were performed for each predator species.

### 2.4. Data Analysis

The hypothesized biological control agents’ relationships were analyzed using the logistic procedures of SAS^®^ (Stockholm, Sweden). Maximum likelihood analyses were conducted on binary response data fitted for logistic models. A binomial logistic regression model was fitted to the data to estimate the maximum likelihood of predation between these predators, mortality of the fall armyworm larvae due to predation and prey preference per predator species. Comparisons were drawn based on the larval mortality of each predator species.

A multinomial logistic regression model was fitted to estimate the likelihood of both predators co-existing within the same guild (=in the absence of competition) [35]. This model allows for the calculation of predictive values (probabilities) that can then be used to discern the possibility of one event occurring over another (S.A.S. Institute 2013) [32]. Similarly, the analysis compared prey preference by predator species on armyworm larvae type (non-parasitized or parasitized). Reference points for the analysis were designated as: time = 20 min and no feeding of any kind (D.N.F.). All statistical analyses were performed using SAS^®^ software (S.A.S. Institute 2013) [36].

## 3. Results

### 3.1. Predator vs. Predator, Intraguild Predation

#### 3.1.1. Without Fall Armyworm

Predation amongst the first (β = −2.16; df = 1; *p* = 0.0008) and second (β = −1.09; df = 1; *p* = 0.0194) nymphal life stages of both pentatomid species was not significant. In contrast, late nymphal life stages—fourth and fifth instar—were heavily preyed and found significant within these competitive pairings, (β = 1.276; df = 1; *p* = 0.0015) and (β = 1.10; df = 1; *p* = 0.0053), respectively. Overall, predation was not uniform across the competitive pairings. It was significantly different from the third instar reference point (Table 2, Figure 1). Similarly, predation across species was not uniform, as predation of *E. floridanus* life stages was less likely to occur (β = −1.52; df = 1; *p* < 0.0001) when compared to *P. maculiventris* life stages. As a result, survivorship of *E. floridanus* life stages was significantly higher than that of *P. maculiventris* life stages.

#### 3.1.2. With Fall Armyworm

A total of 60 fall armyworm larvae were provided to the pentatomid predators. Observed feeding behavior revealed that with the introduction of the fall armyworm, no feeding of any kind was more likely to occur than any other hypothesized feeding behavior (Table 3). As for the remaining behaviors, solitary predation (only one pentatomid fed on the fall armyworm) and intraguild predation (one predator fed on the pentatomid) were twice as likely to occur (β_1_ = −0.59; df = 1; *p* = 0.2920 and β_2_ = −0.59; df = 1; *p* = 0.2920, respectively) than both pentatomids sharing the fall armyworm larva (β = −2.20; df = 1; *p* = 0.0371) (both fed on the fall armyworm) (Table 3).

### 3.2. Predator vs. Parasitoid

When parasitized larvae of fall armyworm (*n* = 20) were provided to *E. floridanus*, 11 larvae were killed and 9 were left alive; similarly, for 20 non-parasitized larvae, 12 were killed and 8 larvae were left alive. Of the 20 parasitized larvae provided to *P. maculiventris*, 11 were killed and 9 larvae were left alive. In contrast, among the non-parasitized fall armyworm larvae were provided to *P. maculiventris*, 17 were killed by the predator and 3 were left alive. Mortality of non-parasitized fall armyworm larvae (β = 0.39; df = 1; *p* = 0.1034) was higher than for parasitized larvae, but the differences were not statistically significant. This indicates that predators were likely to feed on either parasitized or non-parasitized larvae. *Podisus*
*maculiventris* displayed a preference for non-parasitized larvae (see also Section 3.3). In addition, fall armyworm mortality was more likely to occur in experiments using *P. maculiventris,* given it was less likely to occur with *E. floridanus* (β = −0.28; df = 1; *p* = 0.2390) (Table 4). Nevertheless, both species could produce the same level of fall armyworm mortality.

### 3.3. Predator–Prey Preference

In choice tests using an olfactometer with *E. floridanus*, no parasitized larvae of fall armyworm (*n* = 20) were chosen; for non-parasitized larvae, 2 were chosen and 18 were not. For similar tests with 20 larvae of *P. maculiventris*, 7 were chosen and 13 were not; for non-parasitized larvae, 13 were chosen and 7 were not. When parasitized larvae of fall armyworm (*n* = 20) were provided in Petri dishes to *E. floridanus*, none were chosen; similarly, for 20 non-parasitized larvae, 2 were chosen and 18 larvae were not. Four of the twenty parasitized larvae provided to *P. maculiventris* were chosen and sixteen were not. Non-parasitized larvae of fall armyworm were provided to *P. maculiventris*, and 8 larvae were chosen and 12 were not. *Euthyrhynchus floridanus* was significantly less likely to decide between parasitized and non-parasitized larvae (β = −1.17; df = 1; *p* = 0.0035) during the olfactory experiment. Nevertheless, the choice of non-parasitized over parasitized fall armyworm larvae (β = 0.18; df = 1; *p* = 0.5474) was found to be more likely. Non-parasitized larvae were more likely to have be chosen by both predator species (Table 5) based on olfaction only. Similar results were obtained with the use of Petri dish arenas (Table 5), as preference was shown for non-parasitized larvae (β = 0.62; df = 1; *p* = 0.0695) and distinct prey choice was less likely with *E. floridanus* (β = −1.09; df = 1; *p* = 0.0076). Predation of the fall armyworm was as expected, despite the lack of differentiation between parasitized and non-parasitized larvae. Both methods produced similar conclusions, as analyses revealed no significant difference (Table 6).

## 4. Discussion

### 4.1. Predator Competition

Predation rates and their efficacy for biological control (as exhibited by predators sharing a guild) may suffer due to the competition between these species [37,38], especially when the common prey source is scarce or at low population numbers. The complex relationship between predators within the same guild is still not known. However, the competitive pairings were established to mimic a new dynamic among hunting generalist predators within a single guild [37,39]. The study confirmed antagonistic behavior amongst two Asopinae species, *Podisus maculiventris* and *Euthyrhynchus floridanus**,* commonly found in the southeast of the United States. In this laboratory setting, adult *E. floridanus* preyed on all offered life stages of *P. maculiventris*. However, this was not consistent when the roles were reversed, given the difference in the mortalities. We believe that *E. floridanus* would be an intraguild predator of *P. maculiventris* within any given guild [40,41].

Nevertheless, the exact reason for neglecting immature instars is unknown. Nymphal agility and evasion ability are good prospects for evaluation. Moreover, prior predatory behavior reports indicate that *P. maculiventris* will feed on small prey, despite needing larger prey to accelerate development [42]. It is, therefore, possible that the results of this experiment were influenced by the nutritional requirements of both species, given that the insects used were recent molts.

When fall armyworm larvae were used in a similar experimental setup, predatory behavior became more complex. Upon release, predators were observed lacking any decision to feed on the fall armyworm and instead circled the arena while trying to avoid each other. When feeding did begin, *E. floridanus* was observed feeding first. Earlier observations reported that a phenomenon of predators deterring their competitors using bodily fluids is possible [27]. In doing so, these predators often faced no competition or disturbance while feeding and hunting. The phenomenon explained by McLain [27] could explain the behavior of both predators once released in the arena. When or how quickly this phenomenon occurs still needs testing. However, with such proximity and the 24 h starvation period before the experiment, the influence of the phenomenon may fade. This became evident throughout the experiment. Initial feeding behavior was not consistent and appeared more fluid with time. According to an earlier study [43], using temporal overlaps may reduce predator antagonism, thereby reducing competition. With the overlapping of the life stages of the predators, there is limited interaction between the life stages of the different species, whether due to size, agility, or dispersal ability. Therefore, they can exist in the same niche. Considering the inconsistency of predation across the different life stages and the impact of time and proximity on feeding behavior displayed under this study, future studies must confirm their roles under open field conditions. The studies will have to evaluate the impact of the differences in life cycles. After all, *P. maculiventris* matures from egg to an adult in as little as 28 days, while *E. floridanus* requires at least 60 days [29,32]. As a result, any augmentative release would eventually lead to interaction between different life stages—for example, an adult *E. floridanus* and *P. maculiventris* nymph.

### 4.2. Prey Preference

Olfactory cues (olfactometer) and direct proximity (Petri dish arena) played a role in both predators’ larval type choice, parasitized and non-parasitized. However, the results were consistent with a study [44] in which chemical (olfactory) cues had a higher response. Similarly, an earlier study confirmed the importance of chemical cues in tracking prey for both *P. maculiventris* and *E. floridanus*; however, chemical (olfactory cues) had a higher response [44]. Despite this, the ultimate choice between parasitized or non-parasitized fall armyworm larvae was more apparent using bodily cues (Petri dish bioassay). It is, therefore, possible that other potential cues (vision, body movements) and chemical cues might have influenced the response seen in the Petri dish arena.

Insect predators have occasionally disrupted parasitism rates in open fields [45]. Nevertheless, generalist predators are needed for biological control to supplement the efforts of specific biological control agents [36] in open fields. In this study, *E. floridanus* predation on the different larval types was less indecisive. Additionally, *P. maculiventris* preferred non-parasitized larvae, similar to earlier reports [46] on pyralid eggs parasitized by *Trichogramma brassicae.* The chemical cues may play a role in identifying parasitized prey. However, as stated earlier, the exact process needs more research in this area.

## 5. Conclusions

The study revealed various feeding behaviors that evolved with time and prey exposure. The competition and aggressive behaviors were apparent between both predators. *E. floridanus* was found to be an intraguild predator of *P. maculiventris*. Intraguild predation was present amongst the biological control agents tested. However, antagonism by competitors was not consistent for all life stages used in this study. When integrating multiple biological control agents, the timing might decide between cooperation and competition between the insect species. However, all three species observed under controlled conditions in this study are present in open fields of maize in Florida. However, their numbers are low. Pest managers seek to improve augmentative biological control to achieve effective I.P.M. of the *S. frugiperda*. Feeding behavior varied with the length of exposure to the food source, and not all predatory life stages of both predators were predated. Finally, the higher mortality seen in non-parasitized fall armyworm larvae for both predator species, despite feeding on parasitized larvae, shows promise and should be further studied. We believe these results can provide insights for biological control practitioners, who should pay attention to these agents while using augmentative biological control for the fall armyworm in Florida. In addition, future studies should target other predators and parasitoids that are critical in different ecological zones where this pest is invading and causing challenges to growers and food industry.

## Figures and Tables

**Figure 1 insects-13-00815-f001:**
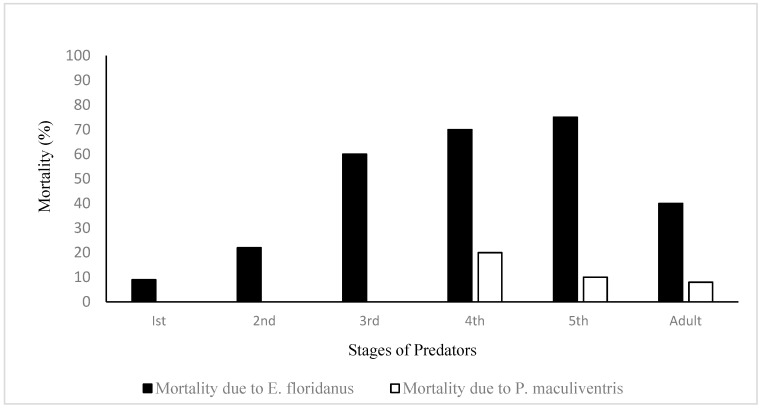
Mortality of two predators’ life stages without fall armyworm offered as prey.

**Table 1 insects-13-00815-t001:** Predation pairings of adults and nymphal instars of *Podisus maculiventris* and *Euthyrhynchus floridanus*.

Life Stages		Life Stages	
*P. maculiventris*	*E. floridanus*	*P. maculiventris*	*E. floridanus*
adult	instar 1	instar 1	adult
adult	instar 2	instar 2	adult
adult	instar 3	instar 3	adult
adult	instar 4	instar 4	adult
adult	instar 5	instar 5	adult
adult	adult	adult	adult

Head capsule widths (*n* = 20) were about 0.35, 0.45, 0.75, 1.3, 2.0, and 2.6 mm, respectively, for instars 1 to 5 and adults, respectively.

**Table 2 insects-13-00815-t002:** Maximum likelihood estimates for all stages of one species being preyed on by adults of the other species.

Variables		Parameter Estimates (β) ± S.E	df	*p*
Species	*E. floridanus*	−1.5184 ± 0.2227	1	<0.0001 ***
	*P. maculiventris*	Reference point		
Life stage				
	First instars	−2.1636 ± 0.6421	1	0.0008 ***
	Second instars	−1.0872 ± 0.4650	1	0.0194 ***
	Third instars	Reference point		
	Fourth instars	1.2705 ± 0.4000	1	0.0015 ***
	Fifth instars	1.1025 ± 0.3952	1	0.0053 ***
	Adult	0.6045 ± 0.3887	1	0.1198 ns

ns = indicates no significant differences (*p* > 0.05). *** indicates highly significant differences (*p* ≤ 0.001).

**Table 3 insects-13-00815-t003:** Likely feeding behavior of *Podisus maculiventris* and *Euthyrhynchus floridanus* for fall armyworm larva ^1^.

Variables	Counts ^2^	Parameter Estimates (β) ± S.E	df	*p*
Observed behavior	60			
One fed alone on the pray	23	−0.5878 ± 0.5578	1	0.2920 ns
Fed on each other	11	−0.5878 ± 0.5578	1	0.2920 ns
No feeding	20	Reference point		
Both fed on prey	6	−2.1972 ± 1.0541	1	0.0371 *

^1^ Likelihood ratio test = 0.2165. Score test = 0.2654. Wald test = 0.3474. ^2^ Observed counts. * = indicates significant differences (*p* ≤ 0.05). ns = indicates no significant differences (*p* > 0.05).

**Table 4 insects-13-00815-t004:** Mortality maximum likelihood estimates of fall armyworm larval type (parasitized and non-parasitized) due to predation.

Variables		Counts ^1^	Parameter Estimates (β) ± S.E.	df	*p*
Species	*E. floridanus*	23	−0.2821 ± 0.2396	1	0.2390 ns
	*P. maculiventris*	28	Reference point		
Treatment (larval type)	Non-parasitized	29	0.3913 ± 0.2403	1	0.1034 ns
	Parasitized	22	Reference point		

^1^ Mortality of fall armyworm. ns = indicates no significant differences (*p*
>
0.05).

**Table 5 insects-13-00815-t005:** Maximum likelihood estimates for fall armyworm larval choice (parasitized or non-parasitized) by predators *Podisus maculiventris* and *Euthyrhynchus floridanus*.

Test Method	Variables		Parameter Estimates (β) ± S.E.	df	*p*
Olfactometer	Species	*E. floridanus*	−1.1677 ± 0.3997	1	0.0035 ***
		*P. maculiventris*	Reference point		
	Treatment (larval type)	Non-parasitized	0.1826 ± 0.3036	1	0.5474 ns
		Parasitized	Reference point		
Petri dish arena	Species	*E. floridanus*	−1.0918 ± 0.4093	1	0.0076 ***
		*P. maculiventris*	Reference point		
	Treatment (larval type)	Non-parasitized	0.6163 ± 0.3395	1	0.0695 ns
		Parasitized	Reference point		

*** = indicates highly significant differences (*p* ≤ 0.001). ns = indicates no significant differences (*p* > 0.05).

**Table 6 insects-13-00815-t006:** Maximum likelihood estimates for olfactometer and Petri dish methods for prey preference study on fall armyworm ^1^.

Variables		Parameter Estimates (β) ± S.E.	df	*p*
Species	*E. floridanus*	−1.1276 ± 0.2855	1	<0.0001 ***
	*P. maculiventris*	Reference point		
Method	Olfactometer	0.0964 ± 0.2198	1	0.6611 ns
	Petri dish	Reference point		
Treatment	Non-parasitized larvae	0.3833 ± 0.2235	1	0.0864 ns
	Parasitized larvae	Reference point		

*** = indicates highly significant differences (*p* ≤ 0.001). ns = indicates no significant differences (*p* > 0.05). ^1^ A total of 160 fall armyworm larvae were tested, and the predators reacted preferentially to 30 of them.

## Data Availability

The data supporting this study’s findings are available from the corresponding author, Muhammad Haseeb, upon reasonable request.

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
