# Peer review of "Intraguild Interactions of Three Biological Control Agents of the Fall Armyworm Spodoptera frugiperda (JE Smith) in Florida"

_insects, 2022, doi:10.3390/insects13090815_

Round 1

Reviewer 1 Report

Line 278. Please delete a come “,”

Line 287-299. Please rewrite this paragraph to improve the understanding, numbers and similar terms combination difficult to get the whole meaning. I would suggest to move part of the content to table 5.

Table 4, 5: Tables are not self-explanatory. I suggest to add more details concern to statistics in the table heading, ie: significant differences are estimated for which variable, and the statistics test used.

Author Response

Comments and Suggestions for Authors

Line 278. Please delete a come “,”

Response: Extra comma was deleted.

Line 287-299. Please rewrite this paragraph to improve the understanding, numbers and similar terms combination difficult to get the whole meaning. I would suggest to move part of the content to table 5.

Response: Agreed, the paragraph was unclear. As suggested, we restructured the paragraph to improve the understanding.

Table 4, 5: Tables are not self-explanatory. I suggest to add more details concern to statistics in the table heading, ie: significant differences are estimated for which variable, and the statistics test used.

Response: In the last column of Table 4, and 5: Nonsignificant, and highly significant results are projected based on the variables and test methods used in this study.

Reviewer 2 Report

The manuscript entitled: “Intraguild Interactions of Three Biological Control Agents of the Fall Armyworm Spodoptera frugiperda (JE Smith) in Florida” submitted for publication in “insects” is considered suitable for publication after minor revision. It gives important information concerning the interactions of the natural enemies of the “fall armyworm”, a serious pest of maize in Florida and in other areas of the world. The results could contribute to the design of effective biological control programs against insect pests.

Some errors are suggested to be revised to the lines of the manuscript presented below:

Line 101: [33] instead of (33]

Line 277: Podisus maculiventris instead of Podisus. Maculiventris

Lines 287-303: all the Latin names should be written in italics

Author Response

Comments and Suggestions for Authors

The manuscript entitled: “Intraguild Interactions of Three Biological Control Agents of the Fall Armyworm Spodoptera frugiperda (JE Smith) in Florida” submitted for publication in “insects” is considered suitable for publication after minor revision. It gives important information concerning the interactions of the natural enemies of the “fall armyworm”, a serious pest of maize in Florida and in other areas of the world. The results could contribute to the design of effective biological control programs against insect pests.

Some errors are suggested to be revised to the lines of the manuscript presented below:

Line 101: [33] instead of (33]

Response: "(" was replaced with "[".

Line 277: Podisus maculiventris instead of Podisus. Maculiventris

Response: Period was removed.

Lines 287-303: all the Latin names should be written in italics

Response: Latin names were Italicized.

Reviewer 3 Report

Review of article:

Intraguild Interactions of Three Biological Control Agents of the Fall Armyworm Spodoptera frugiperda (JE Smith) in Florida

Overview

This article seems hastily written to make it into a special issue. With the information presented, it is difficult to determine the significance of this work. There’s a lot missing in the context of the specific system as well as in the background of the general practice of using similar behavioral trials. There is confusion of basic terms throughout.

The evidence presented implies that the insects’ behavior appeared to be affected by the experimental setups. There was no attempt to ground-truth the results with any field data.

The paper is poorly written with many errors of grammar and structure, and many unclear statements. I have pointed out some, but not all of the unclear statements below. The literature cited tends to be dated (old) and the Discussion section fails to move forward in thinking about these problems. The Discussion is mostly a summary of the results with little thought put into it, and it lacks clarity.

The basis of any Discussion is to focus on which results were statistically significant and explain them clearly. For example, (based on figures and Table 2) you might say, “adults of Species X fed on all nymphal instars of Species Y at a significantly higher rate than when the species’ predatory roles were reversed, however, there was no difference in intraguild predation between adults.” Please explain the significant result as well as the direction of the trend or difference. Then you go on to discuss the relevance and put it into context of what is already known.

Intraguild predation in insects is a well-studied phenomenon. A quick search of my library website using “intraguild predation” + “insects” yielded over 400 recent articles (within the last five years). A good literature review should be able to locate some papers that could lend support to these methods by showing that they can help to predict field behavior/outcomes, or if they cannot, why are you using them? Check out this cool paper I happened to find:  BioControl (2020) 65:297–304; https://doi.org/10.1007/s10526-020-10005-2

It would be helpful to discuss possible improvements to the lab assays for future studies, and to make more specific recommendations for follow-up to this study.

I recommend major revisions to improve these shortcomings. Ideally there would be some field data or references to field data in a similar system. At minimum, I would like to see improvements in the language, the presentation of the existing data, and more thought and background research put into the Discussion. I ran out of time to keep making specific comments in the Discussion, it just needs a major rewrite.

Detailed comments (line by line)

18. Remove “to great lengths” because it is unnecessary and doesn’t make sense

19. I don’t think that “consistent” is the word you’re looking for. Perhaps “frequent” or “regular”?

23. “Indeed” does not make sense here. I think you want “However”

26. use “worldwide” instead of “in the world.”

28. again “consistent” is an odd word choice and so is “facilitated”

31. Introduce your agents first then talk about the investigations

34-37. Poor sentence structure (two verbs)

38. Remove “aimed to investigate…” because these are just extra words that don’t add meaning

31-32. Unclear sentence beginning with “The specific investigations…”

29-31. I am not sure that your paper made any recommendations as to “how we can possibly integrate…”

49. by Poaceae do you mean turf grass?

66-82. Somewhere in this paragraph can you give a very brief explanation of how these are used in biocontrol programs? On line 79 it says they are “commonly found,” what does that mean? Are they established populations from earlier releases? Later (in Discussion) you say that their “numbers are low.” Which is it? Common or not?

81. Remove “frequent” because it doesn’t make sense in combination with “seldom.” Or replace “frequent” with another word such as “efficient” or “adequate.”

81-82. This last sentence is vague. We could use a whole paragraph here outlining the different pairings and why they were chosen. Include why 3rd instar FAW are chosen as prey and why 2nd instar nymphs were chosen for prey preference testing.

86. remove the comma

101. Methods described “earlier” usually means within the same paper but you are referring to an earlier paper. Also bracket incorrectly used.

98. If R. Meagher is an author on this paper is it appropriate to cite them as giving “personal communication?”

106. what does “newly emerged” mean when applied to larvae?

117. One adult, but how many nymphs? Assuming one, but this should be stated.

121-122. “Also..” this whole sentence is not needed

142. one predator fed on the OTHER predator, or on the other Pentatomid

142-144. I don’t understand how these controls were used in the analysis. This is not discussed in section 3.1.2.

161. The statement “All larvae were confirmed dead..” belongs in the results, not the methods.

164-165. Sentence fragment beginning with “An olfactory method…”  Also the word “choice” should appear somewhere in this paragraph to help describe the setup.

168-169. The control setup is not described clearly. I don’t understand if the controls offered a different choice, between a non-parasitized FAW and the other Pentatomid species nymph?  Or was it a choice between the n-p FAW and nothing?

194. If larvae are moving freely in the dish, how can you tell which one is parasitized?

201. Remove “Moreover,” this word is not helpful here.

206-213. Is “guild void” a real term? I don’t understand this sentence. If they are coexisting how can the guild be void?  It would be nice to have some field data to validate any of this. For example count some plots and see what the relative frequencies and life stages of the bugs are. Do they actually coexist, or just potentially?

211-212. Why is the word “however” used here? This is used for drawing a contrast, which is not what you are doing. I also don’t understand the reference points used. Are you trying to say that if a trial had no feeding after 20 minutes, it was put into the reference group? 

215. Predator vs. Predator here is not “competition”…  it’s intraguild predation

217. Mortality due to predation?  Why not say predation or “feeding on?” If they all eventually died of attempted predation, why do you want to call it mortality instead of predation? Also, in the figures you present survivorship, not mortality. Avoid switching back and forth between these, please be consistent if you are going to use them.

228-231. It would be more helpful to integrate these counts into Table 3 rather than stating them here.

233. twice as likely, not twice as MORE likely

234.  Why is “compete” defined when one predator fed on the other predator? That’s not a standard definition of competition. That is intraguild predation.

237, 254. Honestly Figures 1 and 2 are not useful. The grouped bars are just presenting reciprocal numbers. It would be much more interesting to compare percent MORTALITY (not survival) of the species together in one figure. The grouped bars would be a bar for each species grouped at each life stage, and the bar height would be percent mortality.  Figure caption “(without prey)” should be “(without FAW offered as prey)” Also please change the bars in the figures to plain bars (without shading) and adjust the colors for color-blind people (more contrast between the colors or use black and white).

270-274. Please present raw numbers in a tabular format.

277. fix scientific name

278. remove extra comma

287-295. Please present the raw numbers in a tabular format.

287-302. Fix scientific names

303-304. Did you mean to say that they were expected? Instead of not expected? Also, instead of “method types” it is more straighforward to just stay “methods.”

317-318. Is FAW “scarce?” Or do you mean during times when it is scarce, while at other times it is not? Is there any information about the timing of the life cycles of all these different insects? How many generations and how long they take to develop? Do they coexist already and if so, what life stages coexist?

332. Specimens are usually dead. Like in a museum. These are test subjects, or just call them insects.

342-345. Entire section is not clear. I don’t even know if you are talking about your experiment or speculating on a potential field condition. What does this mean? “Initial feeding behavior was not consistent and appeared more fluid with time.”

356. What does “overlapping the life stages” mean?

370. “Based on the results” is understood and should not be stated.

Author Response

Comments and Suggestions for Authors

Review of article:

Intraguild Interactions of Three Biological Control Agents of the Fall Armyworm Spodoptera frugiperda (JE Smith) in Florida

Overview

This article seems hastily written to make it into a special issue. With the information presented, it is difficult to determine the significance of this work. There’s a lot missing in the context of the specific system as well as in the background of the general practice of using similar behavioral trials. There is confusion of basic terms throughout.

The evidence presented implies that the insects’ behavior appeared to be affected by the experimental setups. There was no attempt to ground-truth the results with any field data.

The paper is poorly written with many errors of grammar and structure, and many unclear statements. I have pointed out some, but not all of the unclear statements below. The literature cited tends to be dated (old) and the Discussion section fails to move forward in thinking about these problems. The Discussion is mostly a summary of the results with little thought put into it, and it lacks clarity.

The basis of any Discussion is to focus on which results were statistically significant and explain them clearly. For example, (based on figures and Table 2) you might say, “adults of Species X fed on all nymphal instars of Species Y at a significantly higher rate than when the species’ predatory roles were reversed, however, there was no difference in intraguild predation between adults.” Please explain the significant result as well as the direction of the trend or difference. Then you go on to discuss the relevance and put it into context of what is already known.

Intraguild predation in insects is a well-studied phenomenon. A quick search of my library website using “intraguild predation” + “insects” yielded over 400 recent articles (within the last five years). A good literature review should be able to locate some papers that could lend support to these methods by showing that they can help to predict field behavior/outcomes, or if they cannot, why are you using them? Check out this cool paper I happened to find:  BioControl (2020) 65:297–304; https://doi.org/10.1007/s10526-020-10005-2

It would be helpful to discuss possible improvements to the lab assays for future studies, and to make more specific recommendations for follow-up to this study.

I recommend major revisions to improve these shortcomings. Ideally there would be some field data or references to field data in a similar system. At minimum, I would like to see improvements in the language, the presentation of the existing data, and more thought and background research put into the Discussion. I ran out of time to keep making specific comments in the Discussion, it just needs a major rewrite.

Detailed comments (line by line)

  1. Remove “to great lengths” because it is unnecessary and doesn’t make sense

Response: “to great lengths” was removed.

  1. I don’t think that “consistent” is the word you’re looking for. Perhaps “frequent” or “regular”?

Response: “Consistent” was replaced with “regular”.

  1. “Indeed” does not make sense here. I think you want “However”

Response: “Indeed” was replaced with “However”.

  1. use “worldwide” instead of “in the world.”

Response: Worldwide was been added.

  1. again “consistent” is an odd word choice and so is “facilitated”

Response: Consistent and facilitated were replaced with Regular and promoted, respectively.

  1. Introduce your agents first then talk about the investigations

Response: Paragraph flow was restructured.

34-37. Poor sentence structure (two verbs)

Response: Sentence format was adjusted.

  1. Remove “aimed to investigate…” because these are just extra words that don’t add meaning

Response: Sentence reworded to the study “investigated”.

31-32. Unclear sentence beginning with “The specific investigations…”

Response: Agreed, the sentence was reworded.

29-31. I am not sure that your paper made any recommendations as to “how we can possibly integrate…”

Response: The paper only presents the outcomes of likely cooperation between the insects. The likelihood for cooperation between the different life stages mean we could overlap the species to create an integration plan.

  1. by Poaceae do you mean turf grass?

Response: Yes, this referred to grasses. The word was replaced with the actual meaning, “turf grass”.

66-82. a) Somewhere in this paragraph can you give a very brief explanation of how these are used in biocontrol programs? b) On line 79 it says they are “commonly found,” what does that mean? c) Are they established populations from earlier releases? d)Later (in Discussion) you say that their “numbers are low.” Which is it? Common or not?

Response: a) Reports on the use of these species in biocontrol programs are lacking, so there are no indication of numbers augmentatively released. Instead reports focus on their feeding efficacy or parasitism rates in open fields, especially Cotesia and P. maculiventris.

  1. b) “Commonly found” refers to their appearance in open fields.
  2. c) No, they are native occurring species that are already part of the ecosystem.
  3. d) The “low numbers” is in response to an estimated amount that would be needed to supplement insecticide use.
  4. Remove “frequent” because it doesn’t make sense in combination with “seldom.” Or replace “frequent” with another word such as “efficient” or “adequate.”

Response: As suggested, the word “frequent” was removed.

81-82. This last sentence is vague. We could use a whole paragraph here outlining the different pairings and why they were chosen. Include why 3rd instar FAW are chosen as prey and why 2nd instar nymphs were chosen for prey preference testing.

Response: Additional information was added to the discussion area

  1. remove the comma

Response: Comma was removed.

  1. Methods described “earlier” usually means within the same paper but you are referring to an earlier paper. Also bracket incorrectly used.

Response: As suggested, the citation was corrected

  1. If R. Meagher is an author on this paper is it appropriate to cite them as giving “personal communication?”

Response: Perhaps not, the citation was modified to include only his Research Assistant.

  1. what does “newly emerged” mean when applied to larvae?

Response: Referring to the newly molted larvae before sclerosis was completed. 

  1. One adult, but how many nymphs? Assuming one, but this should be stated.

Response: More information was added in Table 1; highlights the pairing per arena.

121-122. “Also.” this whole sentence is not needed

Response: The sentence was deleted.

  1. one predator fed on the OTHER predator, or on the other Pentatomid

Response: Predator has been replaced with Pentatomid for clarity

142-144. I don’t understand how these controls were used in the analysis. This is not discussed in section 3.1.2.

Response: The controls were used to verify feeding by the Pentatomids on the fall armyworm. In the analysis, the likelihood comparisons were refer to an actual feeding pairing. This was done to discern the differences in each pairing at an active feeding stage.

  1. The statement “All larvae were confirmed dead..” belongs in the results, not the methods.

Response: The statement has been moved to an appropriate location

164-165. Sentence fragment beginning with “An olfactory method…”  Also the word “choice” should appear somewhere in this paragraph to help describe the setup.

Response: As suggested, the sentence was formatted and choice was added to improve clarity

168-169. The control setup is not described clearly. I don’t understand if the controls offered a different choice, between a non-parasitized FAW and the other Pentatomid species nymph?  Or was it a choice between the n-p FAW and nothing?

Response: The choice was between n-p FAW and nothing. The sentence was restructured to clarify this.

  1. If larvae are moving freely in the dish, how can you tell which one is parasitized?

Response: The Petri dish was observed actively for 10 mins. There was also little movement from the parasitized FAW during this time, which made it easier to distinguish between the two.

  1. Remove “Moreover,” this word is not helpful here.

Response: “Moreover” was removed.

206-213. Is “guild void” a real term? I don’t understand this sentence. If they are coexisting how can the guild be void?  It would be nice to have some field data to validate any of this. For example count some plots and see what the relative frequencies and life stages of the bugs are. Do they actually coexist, or just potentially?

Response: “Void” was a typo and has been removed.

211-212. Why is the word “however” used here? This is used for drawing a contrast, which is not what you are doing. I also don’t understand the reference points used. Are you trying to say that if a trial had no feeding after 20 minutes, it was put into the reference group?

Response: After 20 minutes predators had decided, so we used this time interval as the basis for when a choice would be made. It served as the reference point in determining the likelihood of the choice being otherwise. Therefore, if no feeding occurred after 20 minutes the statistical inferences show that it was unlikely that would change.

  1. Predator vs. Predator here is not “competition”… it’s intraguild predation

Response: Agreed, the correct term has been used.

  1. Mortality due to predation? Why not say predation or “feeding on?” If they all eventually died of attempted predation, why do you want to call it mortality instead of predation? Also, in the figures you present survivorship, not mortality. Avoid switching back and forth between these, please be consistent if you are going to use them.

Response: Agreed, mortality has been replaced with predation for clarity and consistency.

228-231. It would be more helpful to integrate these counts into Table 3 rather than stating them here.

Response: The counts have been integrated to table 3.

  1. twice as likely, not twice as MORE likely

Response: Agreed, “more” has been removed.

  1. Why is “compete” defined when one predator fed on the other predator? That’s not a standard definition of competition. That is intraguild predation.

Response: Agreed, the terminology has been corrected.

237, 254. Honestly Figures 1 and 2 are not useful. The grouped bars are just presenting reciprocal numbers. It would be much more interesting to compare percent MORTALITY (not survival) of the species together in one figure. The grouped bars would be a bar for each species grouped at each life stage, and the bar height would be percent mortality.  Figure caption “(without prey)” should be “(without FAW offered as prey)” Also please change the bars in the figures to plain bars (without shading) and adjust the colors for color-blind people (more contrast between the colors or use black and white).

Response: As suggested both figures were merged into one based on the percent mortality.

270-274. Please present raw numbers in a tabular format.

Response: A “Counts” column has been added per an earlier suggestion from this reviewer.

  1. fix scientific name

Response: “Podisus Maculiventris” has been changed to “Podisus maculiventris”.

  1. remove extra comma

Response: Extra comma removed.

287-295. Please present the raw numbers in a tabular format.

Response: A “Counts” column has been added.

287-302. Fix scientific names

Response: Scientific names have been fixed.

303-304. Did you mean to say that they were expected? Instead of not expected? Also, instead of “method types” it is more straightforward to just stay “methods.”

Response: Although predation on the fall armyworm was expected, highlighting the non-significance of parasitism in larval choice was the intention.

“Method types” has been reduced to “methods”.

317-318. a)Is FAW “scarce?” Or do you mean during times when it is scarce, while at other times it is not? b)Is there any information about the timing of the life cycles of all these different insects? c)How many generations and how long they take to develop? d)Do they coexist already and if so, what life stages coexist?

Response: a) The sentence refers to periods of low populations for the prey species.

b & c) Unfortunately, information on lifecycle timing is separate for each species and does not detail periods of overlap between generations of each species. Therefore, these deductions have to be made based on life history articles that show active seasons. Similarly, field sampling when scouting fall armyworm damage (just observations, no recorded data) has resulted in their collection within the same season.

Podisus  maculiventris takes approximately 28 – 40 days from egg to adult and is most active from spring to fall with 2-3 annual generations. Euthyrhynchus floridanus takes about 60 days minimum from egg to adult. Populations of E. floridanus tend to peak in spring and fall but it can be found year-round. Generations vary from 1 – 2 per year. Cotesia marginiventris completes its life cycle in  approximately 15 days and is active during the summer. The overlapping presence of these all these insects would be spring into summer.

  1. d) To our knowledge, this is the first study to propose the coexistence of these species in this manner. Based on our report, pentatomid adults and younger instars are capable of coexisting. The parasitoid larvae however, would fall victim to both pentatomids.
  2. Specimens are usually dead. Like in a museum. These are test subjects, or just call them insects.

Response: Agreed, the terms has been replaced with insects.

342-345. Entire section is not clear. I don’t even know if you are talking about your experiment or speculating on a potential field condition. What does this mean? “Initial feeding behavior was not consistent and appeared more fluid with time.”

Response: Feeding behavior at the beginning stages of the experiment were not always the same at the end of the experiment. Probing might have evolved to feeding or feeding might have for no other reason. The section has been adjusted for clarity.

  1. What does “overlapping the life stages” mean?

Response: Refers to the multiplying life stages of the species simultaneously existing. The sentence has been rewritten for clarity.

  1. “Based on the results” is understood and should not be stated.

Response: As suggested, the statement has been removed and the sentence has been rephrased.

Round 2

Reviewer 3 Report

The manuscript has been improved significantly, however, many typos remain that hopefully will be caught by final editorial processing.

Table 1 footnote: do both species have the same head capsule measurements? So you are saying that they are approximately the same size? This would make a useful sentence in the introduction. Why bury it in a footnote?

The new figure is better, but needs a y-axis label and the legend (or caption) could be improved to clarify, for example, "Mortality of P. maculiventris due to E. floridanus."

lines 411-413: They did not present ANY data on the preference of predators for different instars of the armyworm. This statement is completely unsupported.

Why is olfactometer being capitalized?

Some of the author's responses to my questions did not make it into the manuscript. It is unclear why they would want to leave out basic information about the predator life cycles. Perhaps among people who study this system the information is unnecessary or too redundant. Perhaps in this special issue the information will be covered elsewhere. Perhaps you don't want to include more citations. But, as an outside reviewer I would prefer to just get the information here, clearly and succinctly. 

Author Response

Reviewer #3 (Round 2)

The manuscript has been improved significantly, however, many typos remain that hopefully will be caught by final editorial processing.

Table 1 footnote: do both species have the same head capsule measurements? So you are saying that they are approximately the same size? This would make a useful sentence in the introduction. Why bury it in a footnote?

Response: No, the footnote referred to the head capsule of the fall armyworm. It was included in table one due to the relevance of table 1 to methodology 2.2.2. However, given the confusion, the specific measurement was added to the appreciate section of 2.2.2. and the foot note remove.

We agree that the discerning measurements are needed for the pentatomid species and have added them to the introduction.

The new figure is better, but needs a y-axis label and the legend (or caption) could be improved to clarify, for example, "Mortality of P. maculiventris due to E. floridanus."

Response: The caption of the figure has been edited to “Pentatomid mortality as a result of predation from the other species without fall armyworm offered as prey”. This legend has been modified.

lines 411-413: They did not present ANY data on the preference of predators for different instars of the armyworm. This statement is completely unsupported.

Response: Agreed, the statement was based on our observations. However, we did not quantify this observations. Therefore, the statement “In addition, this study revealed that both species of predators preferred late larval instars of the fall armyworm” has been deleted.

Why is olfactometer being capitalized?

Response: With the exception of the heading “2.3.1. Olfactometer” or the word starting a sentence, the remaining instance have been changed to a lowercase “o”.

Some of the author's responses to my questions did not make it into the manuscript. It is unclear why they would want to leave out basic information about the predator life cycles. Perhaps among people who study this system the information is unnecessary or too redundant. Perhaps in this special issue the information will be covered elsewhere. Perhaps you don't want to include more citations. But, as an outside reviewer I would prefer to just get the information here, clearly and succinctly. 

Response:  There are far more detailed reviews that would make our inclusion of a similar life history redundant. Nevertheless, we understand the need for an article that covers everything and have included some information on the length of the life cycle in the discussion. We have also cited a few studies in the narrative. 
